# Biological Background of Resistance to Current Standards of Care in Multiple Myeloma

**DOI:** 10.3390/cells8111432

**Published:** 2019-11-13

**Authors:** Pedro Mogollón, Andrea Díaz-Tejedor, Esperanza M. Algarín, Teresa Paíno, Mercedes Garayoa, Enrique M. Ocio

**Affiliations:** 1Hospital Universitario de Salamanca (IBSAL), Centro de Investigación del Cáncer-IBMCC (CSIC-USAL), 37007 Salamanca, Spain; pmog@usal.es (P.M.); andreadiaz0411@gmail.com (A.D.-T.); macalgpac@gmail.com (E.M.A.); tpaino@usal.es (T.P.); mgarayoa@usal.es (M.G.); 2Hospital Universitario Marqués de Valdecilla (IDIVAL), Universidad de Cantabria, 39008 Santander, Spain

**Keywords:** multiple myeloma, resistance, proteasome inhibitors, immunomodulatory agents, monoclonal antibodies, resensitization

## Abstract

A high priority problem in multiple myeloma (MM) management is the development of resistance to administered therapies, with most myeloma patients facing successively shorter periods of response and relapse. Herewith, we review the current knowledge on the mechanisms of resistance to the standard backbones in MM treatment: proteasome inhibitors (PIs), immunomodulatory agents (IMiDs), and monoclonal antibodies (mAbs). In some cases, strategies to overcome resistance have been discerned, and an effort should be made to evaluate whether resensitization to these agents is feasible in the clinical setting. Additionally, at a time in which we are moving towards precision medicine in MM, it is equally important to identify reliable and accurate biomarkers of sensitivity/refractoriness to these main therapeutic agents with the goal of having more efficacious treatments and, if possible, prevent the development of relapse.

## 1. Introduction

Treatment of multiple myeloma (MM) has experienced a revolution in recent years that has resulted in a significant improvement in the outcome of these patients, particularly the younger ones, with at least a doubling of median overall survival (OS) [1]. Alkylators and steroids have remained the backbone for MM treatment for many decades [2] until the beginning of the current century when immunomodulatory drugs (IMiDs) (thalidomide, lenalidomide, and pomalidomide), proteasome inhibitors (PIs) (bortezomib (BTZ), carfilzomib (CFZ), and ixazomib (IXZ)), and anti-CD38 monoclonal antibodies (mAbs) (daratumumab and probably soon isatuximab) were approved and became the backbone of the current standards of therapy both in the USA [3] and Europe [4]. Moreover, research looking for new therapeutic targets is still very active in MM, with several novel agents knocking at the door, such as Bcl-2 inhibitors (venetoclax), Exportin-1 (XPO-1) inhibitors (selinexor, recently approved by the FDA), or promising immunotherapeutic approaches including antibody-drug conjugates, bispecific T cell engagers (BiTEs), or CAR-T cells, mainly directed against BCMA and/or other plasma cell-specific antigens [5].

Despite all these advances, for the majority of patients, MM is an incurable disease, mainly due to the presence of resistance by tumor plasma cells to the specific administered treatments. The development of resistance is evidently associated with a bad prognosis, and, in fact, in a study performed some years ago, the median OS of patients developing resistance to both BTZ and lenalidomide was of only 9 months [6]. Moreover, the recent concept of penta-refractory patients (refractory to BTZ, CFZ, lenalidomide, pomalidomide, and an anti-CD38 mAb) is considered an unmet medical need with progression-free survival (PFS) in different daratumumab-based combinations of approximately 2 months [7]. In this same line, the recently FDA-approved XPO-1 inhibitor selinexor, provided a 20% overall response rate (ORR), with PFS and OS of only 2.3 months and 5.5 months, respectively, in these highly refractory patients [8,9].

This resistance can be either primary, represented by a lack of response to the administered treatment, or secondary, when patients progress after an initial period of response. These two types of resistance are assumed to have arisen through different mechanisms: the primary one can be derived from intrinsic abnormalities or features inherently present in the tumor cell, while the second one, likely represents adaptive survival mechanisms of tumor cells to the treatment to which they are being exposed. Another possibility for this secondary resistance is the event of clonal selection that relies on the existence of clonal heterogeneity within the tumor. Recent work has demonstrated a clonal competition for dominance among different clones [10]. Spontaneity or therapeutic pressure may lead to the selection of intrinsically resistant minority clones initially present within the bulk of the tumor population. In addition, spatial heterogeneity has also been demonstrated in MM, where different focal lesions (FLs) show distinct genomic profiles, which may lead to the development of resistance in different niches in the bone marrow [11].

Another factor to be considered is the role of the microenvironment in the development of resistance, and this is true not only in the case of immunotherapeutic approaches but also for other targeted strategies. Interactions of myeloma plasma cells with components of the bone marrow microenvironment (e.g., extracellular matrix, mesenchymal stromal cells (MSCs), osteoclasts, and immune cells), mediated by direct contact or by soluble factors, have been widely recognized as key determinants in myeloma pathogenesis [12]. Moreover, increasing evidence shows the involvement of extracellular vesicles (EVs), including exosomes, in several aspects of myeloma progression and in enabling the acquisition of drug resistance [13,14].

Independently from the type of resistance, it is of utmost importance to deepen research into responsible mechanisms, in order to design strategies that could avoid the acquisition of resistance or overcome it. In this manuscript, we review the knowledge on the mechanisms underlying resistance to the current backbones in MM treatment constituted by PIs, ImiDs, and mAbs. For this purpose, we first briefly explain the mechanisms of action for each of these families of agents; subsequently we analyze the potential mechanisms involved in the development of resistance based on preclinical or clinical observations, and, finally, strategies that could overcome these mechanisms and may be used in the clinical setting are proposed.

## 2. Resistance to Proteasome Inhibitors

Myeloma plasma cells seem to be especially sensitive to the activity of PIs, due to their dependence on the Ubiquitin-Proteasome System (UPS) for the processing of defective immunoglobulins (Igs) and other proteins [15,16]. In fact, the incorporation of the first in class PI, BTZ, to the anti-myeloma armamentarium in 2003 can be considered one of the major milestones in the treatment of myeloma patients, because its use as a single agent or in combinatorial regimens has become a backbone both in the front-line and relapsed/refractory settings [17]. Subsequently, second-generation PIs structurally different to BTZ were developed with the aim of improving BTZ efficacy and its safety profile (majorly avoiding peripheral neuropathy), as well as overcoming BTZ resistance [17,18].

### 2.1. Mechanisms of Action

BTZ and ixazomib (IXZ; MLN9708) are dipeptide boronic acid derivatives, which, by covalent binding, reversibly and preferentially inhibit the β5 subunit of the proteasome and the β5i subunit of the immunoproteasome. They also bind to a lesser extent, and with a much lower affinity, to the β1 subunits [19,20].

The already approved carfilzomib (CFZ; PR171) and its oral structural analog, oprozomib (OPZ; ONX0912), are epoxyketones that specifically and irreversibly bind and inhibit the β5 subunits of the constitutive proteasome and the β5i subunits of the immunoproteasome [21,22]. Other second-generation PIs in clinical development are delanzomib, a reversible boronate also primarily inhibiting the β5 subunit of the proteasome [23], and marizomib (salinosporamide A/NPI-0052), a naturally produced β-lactone, which irreversibly inhibits the β5, β2, and β1 subunits of the proteasome [24].

Broad effects have been reported after PI treatment in MM (e.g., blockade of NF-κB activation, cell cycle arrest, intrinsic and extrinsic induction of apoptosis, inhibition of DNA repair enzymes, and the inhibition of adhesion of myeloma and bone marrow stromal cells) (reviewed in [25,26,27]), which likely contributes to the clinical efficacy of these agents in MM. In addition, BTZ [28,29,30], IXZ [31], CFZ, and OPZ [32] have been reported to have bone anabolic and anti-resorptive effects, rendering a beneficial effect on myeloma bone disease.

### 2.2. Mechanisms of Resistance

Despite the important anti-MM activity of PIs, both primary and secondary resistance to these agents is quite common. The initially proposed resistance mechanisms to these agents, based on MM cell line models with acquired resistance to PIs, were genomic or functional abnormalities in the proteasome subunits. Mutations at the PI-binding pocket of the β5 proteasome subunit (*PSMB5)*, such as Thr21Ala, Ala49Thr, and Ala50Val, or in its proximity (Cys52Phe, Met45Ile, and Met45Val), cause distinct levels of BTZ resistance [33,34,35]. However, these *PSMB5* point mutations are really infrequent in patients (0% at diagnosis and 1% in relapsed and refractory (RRMM)) [36]. In addition to the *PSMB5* mutations, resistant MM cell lines have frequently been found to overexpress the β5, β2, and β1 subunits of the proteasome, usually accompanied by increased catalytic chymotrypsin, trypsin, and caspase-like activity, respectively, and subsequent higher cellular survival rates as compared to sensitive cell lines [37,38,39]. In this same line, Wang’s group reported higher β5 expression in a BTZ-resistant MM patient when compared to sensitive patients [40]. Sometimes both mechanisms are found together: cells harboring mutations in *PSMB5* overexpress its mutant and structurally altered β5 subunit [35], thereby leading to higher resistance to PIs in MM cell lines. Another mechanism involved in BTZ and CFZ resistance, and closely related to the previous ones, is the overexpression, through the transcriptional activation of the nuclear factor (erythroid-derived 2)-like (NRF2), of the proteasome maturation protein (POMP) or proteassemblin, a protein involved in the addition of active β-subunits to the proteasome and thus essential for its de novo synthesis [41]. Finally, the proteasome subunit PSMC6, a component of the 19S regulatory particles of the proteasome involved in the ATP-dependent unfolding of substrates and their translocation into the 20S core proteasome, has been shown to be required for BTZ sensitivity in MM cells. In this line, CRISPR-based studies evidenced that deficiency of PSMC6 in the regulatory subunits conferred BTZ resistance by reducing the ability of BTZ to suppress the chymotrypsin-like activity of PSMB5 [42].

Since protein homeostasis in myeloma plasma cells critically depends on the adequate activation of the unfolded protein response (UPR), alterations in UPR/ER-stress proteins are also associated with BTZ resistance. The X-box binding protein 1 (Xbp1) is a transcription factor required for plasma cell differentiation, which also acts as a regulator of the UPR/ER-stress pathway. The active spliced form of Xbp1 (Xbp1s) is commonly downregulated in refractory patients and resistant cell lines [43,44] and has been associated with a de-differentiated status of myeloma cells [44]. *XBP1* inactivating mutations have also been documented in MM patients, promoting BTZ resistance [45]. Besides, the over-expression of heat shock proteins (HSPs) and induction of autophagy are mechanisms by which MM cells may alternatively deal with the increased protein workload generated by PIs and subsequently escape from cell death [46]. The most frequently upregulated HSPs in RRMM are Grp78, HSP90, HSP70, and HSPB8 [47]. Regarding autophagy, the autophagy-inducer Activating Transcription Factor 4 (ATF4) is overexpressed upon proteasome inhibition. Stabilization of ATF4 activates this mechanism through the up-regulation of LC3BII, protecting cells from BTZ-induced death [48]. In line with these mechanisms, Histone Deacetylase 6 (HDAC6) was found to mediate the transport of misfolded proteins to aggresomes, which then transfer protein aggregates to lysosomes for protein clearance via autophagy. The blockade of this mechanism by HDAC inhibitors synergizes with BTZ in MM preclinical models [49,50] and led to the approval of the combination of panobinostat with BTZ and dexamethasone [51]. Additionally, in these UPR mechanisms, increased levels of deubiquitinating enzymes have also been documented to reduce stress levels and promote MM cell survival, thus contributing to PI resistance [52].

Other general mechanisms, not only restricted to proteasome inhibition have also been described. For example, the overexpression of the multidrug efflux transporter MRD1/P-glycoprotein (ABCB1/Pgp) has particularly been associated with resistance to epoxyketone-based PIs [53]. In relation to the bone marrow microenvironment-mediated resistance, direct interaction of myeloma cells and MSCs and MSC-derived IL-6 have been found to partially mediate resistance to BTZ and other PIs [54,55]. Concerning the role of extracellular vesicles (EVs), BTZ-resistant leukemia cells have been shown to overcome proteolytic stress by exocytosis of EVs containing ubiquitinated proteins [56]. In MM, bone marrow MSC-derived EVs have also been shown to induce resistance to BTZ [14]. This resistance is mediated, at least in part, by the transference of proteasome subunit α7 lncRNA (PSMA3-AS1) by MSC-derived EVs to myeloma cells [57]. The main mechanisms of resistance to PIs have been depicted in Figure 1.

### 2.3. Overcoming Drug Resistance

The understanding of the mechanisms of resistance to PIs has also allowed the design of several approaches to overcome it. Thus, the combination of the NRF2 inhibitor all-trans-retinoic acid (ATRA) with BTZ re-sensitizes MM cells resistant to BTZ in preclinical models, suggesting a possible dependency on an increased de novo biosynthesis of proteasomes [41]. Moreover, treatment with 2-methoxyestradiol (2-ME2) and ATRA induces plasma cell differentiation and overcomes myeloma cell resistance to BTZ [58].

Other approaches regarding intrinsic mechanisms of myeloma cells to manage the increased protein workload generated by PIs have been evaluated. Studies have shown that the autophagy inhibitor Chloroquine A overcomes PI resistance and synergizes with PIs [59], being this combination under clinical evaluation in RRMM patients (NCT01438177). Both preclinical and clinical data have demonstrated that HDAC inhibitors synergize with BTZ and overcome resistance, a strategy that has also been clinically approved [49,60]. Likewise, targeting deubiquitinase activity with a novel small molecule inhibitor against USP7 (P5091) induces apoptosis and impairs BTZ resistance in MM [52,61].

In relation to the bone marrow microenvironment, several strategies have been proposed to enhance the sensitivity of MM cells to PIs, such as the prevention of cell adhesion-mediated drug resistance (e.g., through CXCR4 inhibition [62]), or blockade of EV-mediated mechanisms by si-PSMA3-AS1 administration [57].

## 3. Resistance to Immunomodulatory Compounds

The first immunomodulatory agent thalidomide and the subsequently developed derivatives lenalidomide (CC-5013) and pomalidomide (CC-4047) have demonstrated important clinical activity in MM, through the inhibition of proliferation and angiogenesis and, maybe more importantly, several immune-modulating effects [63,64,65,66].

In spite of their chemical similarity, showing only differences within the glutarimide portion, IMiDs differ with respect to several clinical and pharmacological properties that include adverse effects, half-life, metabolism, and clearance [67,68,69,70,71].

In an effort to extend the repository of clinically available immunomodulatory agents, new IMIDs, known as CELMoDs (CRBN modulating agents) have been recently developed. Avadomide (CC-122), closely related to pomalidomide, has shown acceptable safety and tolerability, along with favorable pharmacokinetics [72]. Also, preliminary results evaluating the novel compound iberdomide (CC-220) have demonstrated favorable efficacy and safety in RRMM patients who failed prior therapies, including lenalidomide and pomalidomide [73].

### 3.1. Mechanisms of Action

Cereblon (CRBN), a common primary target of IMiDs, is a substrate recognition component of a DCX (DDB1-CUL4-X-box) E3 protein ligase mediating the ubiquitination and consequent proteasome degradation of target proteins [74]. The binding of IMiDs to CRBN modulates the E3 ligase complex substrate specificity, thereby modifying proteins to be ubiquitinated and degraded by the ubiquitin-proteasome system. Among the several CRBN-binding proteins identified, two of the most down-regulated after IMiD treatment are the lymphocyte transcription factors IKZF1 (Ikaros) and IKZF3 (Aiolos) [75].

Both CRBN expression and IKZF1 and IKZF3 down-regulation are key factors required for the cytotoxic effect of IMiDs. Downstream effects of IMiD-induced CRBN modification include a decrease in the levels of interferon regulatory factor (IRF4) and its target gene *MYC*, an induction of cell cycle arrest by the up-regulation of the cyclin-dependent kinase inhibitor p21^WAF-15^, the promotion of apoptosis mediated by FasL and TRAIL, and the downregulation of the translation checkpoint eIF4E [76,77,78,79].

Interestingly, IKZF1 and IKZF3 degradation occurs within hours after IMiD treatment but it takes another 24–72h to down-regulate IRF4 and c-Myc [75], which suggests that additional mechanisms may be involved in achieving a complete drug response. In this line, the cytotoxic effect of IMiDs has also been associated with an inhibition of Phosphatidil-Inositol-3-Kinase (PI3K-AKT) and Nuclear Factor-κB (NF-κB) pathways [70,80]. Additional effects on immune activation including the stimulation of natural killer and T cells, up-regulation of IL-2 in T cells, and the inhibition of the production of IL-6 and tumor necrosis factor alpha (TNFα) in peripheral blood mononuclear cells (PBMCs) have also been documented [81,82,83].

Structural studies have provided new insights to the understanding of the physical relationship between IMiDs and CRBN, as well as its interaction with substrates. Although thalidomide shows similar binding modes and affinity for CRBN, it is less efficient in targeting IKZF1/3 for degradation by the CRL4^CRBN^ ubiquitin ligase than lenalidomide and pomalidomide. This may be due to a major structural difference between thalidomide and its derivatives, that lies in the presence of a solvent-exposed C4 aniline shared by the latter ones that increases their ability to degrade IKZF1 [84].

The Casein kinase 1 alpha (CK1α) is also a CRBN-binding protein located in chromosome 5q, which is degraded upon IMiD treatment. This degradation results in the activation of p53, which explains the efficacy of lenalidomide in 5q- syndrome [85]. CK1α also has a role in MM pathogenesis as it sustains oncogenic cascades, such as PI3K/AKT and NF-κB [86,87] and has been associated with the modulation of the interferon pathway, TNF-signaling and pro-survival autophagy [88,89].

Apart from the CRBN function as a substrate-receptor of the CRL4 E3 ubiquitin ligase complex, CRBN shares a chaperone-like function promoting the maturation of the protein basigin (BSG). BSG forms a complex with MCT1 (BSG-MCT1) that promotes proliferation, angiogenesis, and invasion. IMiDs compete with BSG for CRBN binding, representing another mechanism for both antitumor activity and teratogenicity [90].

### 3.2. Mechanisms of Resistance

CRBN down-regulation was the first described mechanism of resistance to IMiDs [91]. Preclinical data show that this event is common in in vitro and in vivo models of acquired resistance to IMiDs [77,92]. Clinical studies have also demonstrated some correlation between CRBN expression and resistance to IMiD therapy [93,94]. However, there is still insufficient evidence to use CRBN expression (either mRNA or protein) as a predictive biomarker.

Little is known about the mechanisms regulating CRBN protein turnover. Liu et al. demonstrated that CSN9 signalosome inactivates the Cullin-RING ubiquitin E3 ligase SCF^Fbxo7^, which targets CRBN, by removing the ubiquitin-like Nedd8 protein. Conversely, loss of function of the CSN9 signalosome activated SCF^Fbxo7^ ubiquitin ligase thus promoting CRBN down-regulation and IMiD resistance [95].

Mutations in *CRBN* at the time of IMiD failure or prior to the exposure to these agents have been described and associated with primary and secondary resistance to these agents. However, these mutations are a rare event in both patient samples and cell lines [96,97].

Regarding mechanisms related to substrate affinity and competition, the overexpression of BSG counteracts IMiD-mediated cytotoxicity by preventing its binding to CRBN [90]. Moreover, the expression of substrates itself may also affect IMiD activity. A recent study using a novel targeted mass spectrometry assay described the ordered substrate degradation triggered by IMiDs (IKZF1, IKZF3, CK1α, ZFP91, RNF166, ZNF692, GSPT1, and GSPT2). The increased expression of one of the substrates that should be degraded later may decrease the degradation of previous ones critical for the anti-myeloma effect of the IMiD, therefore leading to resistance [98]. Very recently, it has been reported that RUNX1 and RUNX3 are capable to interact with IKZF1 and IKZF3, protecting them from the CRBN-dependent ubiquitination and degradation induced by IMiDs [99].

Activation of several signaling pathways has also been described to promote IMiD resistance. This is the case of Wnt/β-catenin [100], MEK/ERK [92], or STAT3 pathways [77]. Finally, a higher antioxidative capacity of MM cells has been linked to lenalidomide resistance [101]. See major mechanisms involved in IMID resistance in Figure 1.

### 3.3. Overcoming Drug Resistance

Interestingly, as pointed out before, not all IMiDs hold the same mechanisms of resistance, as in vivo studies from our group suggest that tumors with acquired resistance to lenalidomide respond to pomalidomide and vice versa, achieving a more potent response with pomalidomide. Moreover, the gene expression profile of lenalidomide-resistant cells is different from those resistant to pomalidomide, pointing out to differential mechanisms of resistance to both drugs [92].

Several strategies have been proposed to overcome IMiDs resistance. In this regard, preclinical data suggests that PB-1-102 (STAT3 inhibitor) and Selumetinib (MEK1/2 inhibitor) can overcome IMiD resistance [77,92], pointing out at the importance of these pathways in resistance to this class of agents. Additionally, since the ubiquitin E3 ligase SCF^Fbxo7^ targets CRBN for degradation, combinations with BTZ, or the neddylation inhibitor MLN4924 have demonstrated an enhanced sensitivity to IMiDs [95]. In relation to RUNX proteins protecting IKZFs from degradation through direct interaction, RUNX inhibition (e.g., AI-10-104) resulted in the sensitization of myeloma cells to IMiDs [99]. Lastly, epigenetic modulation has also demonstrated efficacy in overcoming both acquired and intrinsic IMiD resistance. Thus, combinations with 5-Azacytidine (DNA methyltransferase inhibitor) and EPZ-6438 (EZH2 inhibitor) have been shown to resensitize MM cells to IMiDs through an unknown CRBN-independent mechanism [102].

## 4. Resistance to Monoclonal Antibodies

In MM, several monoclonal antibodies (mAbs) are currently being developed in different preclinical and clinical studies. The general mechanism of action of mAbs includes direct effects (apoptosis), immune-mediated cytotoxicity (e.g., antibody-dependent cell-mediated cytotoxicity or ADCC, complement-dependent cytotoxicity or CDC, and antibody-dependent cellular phagocytosis or ADCP) as well as immunomodulatory effects. The approval in 2015 of two mAbs, elotuzumab and daratumumab, initiated the new era of immunotherapy in MM, which is still under active development with the investigation of new mAbs and mAb-based combinations [103].

### 4.1. Mechanisms of Action

Elotuzumab is a humanized IgG1 mAb targeting the extracellular domain of the signaling lymphocytic activation molecule F7 (SLAMF7), also known as CS1. Elotuzumab was already approved in combination with lenalidomide and dexamethasone [104,105], and, most recently with pomalidomide and dexamethasone [106]. This mAb inhibits MM cell adhesion to bone marrow stromal cells and also exerts ADCC [107]. In addition, elotuzumab directly induces the activation of natural killer (NK) cells by binding to SLAMF7 expressed on these type of cells [108].

Daratumumab is a human IgG1 mAb that targets CD38, a cell surface protein that is expressed on MM cells. The proven safety and efficacy of daratumumab has led to its approval both as monotherapy and in combination with standard-of-care treatments, for both RRMM patients and patients with newly diagnosed myeloma [109]. Daratumumab has been described to induce ADCC, CDC, ADCP, and apoptosis via cross-linking [110,111,112] to exert its cytotoxic activity. In addition, daratumumab induces immunomodulatory effects by depleting CD38+ immunosuppressive regulatory T cells (Tregs) and increasing the number, the activation and the clonality of cytotoxic T-cells [113]. Apart from daratumumab, other anti-CD38 mAbs, such as isatuximab (SAR650984), MOR202, and TAK-079 are currently being evaluated in preclinical studies and clinical trials. Isatuximab is a humanized IgG1 anti-CD38 mAb with CDC, ADCC, and ADCP activity [114]. It has been described that it suppresses the induction and function of Tregs [115], although other authors have shown that Tregs are not depleted by this mAb [116]. Isatuximab induces a potent pro-apoptotic activity in the absence of cross-linking agents, together with a strong inhibition of CD38 enzymatic activity [114], and it also triggers lysosome-mediated non-apoptotic cell killing pathways [117]. Isatuximab combined with lenalidomide and dexamethasone [118] or pomalidomide and dexamethasone [119] is active and well-tolerated in RRMM patients. Regarding MOR202 and TAK-079, information about their efficacy and mechanism of action is still very limited. MOR202 is a human IgG1 anti-CD38 mAb, which induces ADCC and ADCP [120,121] but has weak CDC activity [122] and does not induce direct apoptosis [123]. A phase I/IIa study of MOR202 alone and in combination with pomalidomide or lenalidomide demonstrated a good safety profile in heavily pretreated RRMM patients, as well as promising preliminary efficacy and long-lasting tumor control [120]. Moreover, it has recently been shown that lenalidomide enhances MOR202-mediated phagocytosis against myeloma cells by restoring the vitamin D pathway [121]. TAK-079 is a human IgG1 mAb with a high affinity for CD38 that is being developed for the treatment of MM and autoimmune diseases [124]; it is currently being evaluated in a phase I/IIa clinical trial administered as a single agent in RRMM patients (NCT03439280).

### 4.2. Mechanisms of Resistance

Due to the recent introduction of mAbs into the therapeutic armamentarium for MM, the mechanisms involved in both intrinsic and acquired resistance to these type of drugs are still poorly understood (see Figure 1). It would be expected that low/absence target expression is one of the potential causes of intrinsic resistance, however, data in this regard are still controversial. In this sense, it has been observed, both by in vitro experiments and in daratumumab-treated patients, that response to this mAb is significantly associated with CD38 basal expression levels on tumor cells [125,126,127]. On the contrary, other authors found that there were no differences in the expression of CD38 before daratumumab administration between responders and non-responders to this mAb [128,129]. Moreno et al. specifically studied the relationship between CD38 expression level and isatuximab-mediated modes of action, showing that CDC and ADCP mechanisms are induced only in CD38^high^ MM cells, while ADCC is exerted in MM cells with a broad range of CD38 expression [116]. In a recent study, Danhof et al. reported that a strong expression of SLAMF-7 on myeloma cells could be a biomarker for an effective elotuzumab-based treatment, although the patient sample size in this study was very limited [130]. Therefore, more studies are needed to clarify whether SLAMF7 and CD38 are reliable biomarkers of the sensitivity to elotuzumab and daratumumab/isatuximab, respectively.

Factors other than antigen expression, such as patient NK-cell count prior to treatment, could also influence the response to mAb-based therapy in MM [130]. Additionally, polymorphisms in FcγRIIIa (CD16a), an NK cell-receptor involved in ADCC that binds the Fc portion of immunoglobulin G, might also be important, although results are not conclusive so far. In a randomized phase II study of Elotuzumab + Bd (BTZ and low dose dexamethasone) versus Bd in RRMM, EBd-treated patients homozygous for the high-affinity FcγRIIIa V allele (VV) showed longer PFS than those who were homozygous for the low-affinity FcγRIIIa F allele (FF) [131]. However, a sub-analysis of PFS based on the CD16a genotype showed no significant differences between VV and FF in ELOQUENT-2 [132]. In another study, Van de Donk et al. reported that FcγR polymorphisms have only a modest impact on the response to daratumumab and PFS, and did not significantly affect OS [133].

In addition to the basal levels of the target as a potential mechanism of intrinsic resistance, the modulation of target expression after treatment with a specific mAb has also been studied as a possible mechanism of acquired resistance, although results are non-conclusive. It has been reported that daratumumab induces the release of CD38 from myeloma cells by microvesicles [134,135]. In addition, the surface expression of CD38 in myeloma cells is reduced after daratumumab exposure due to the transfer of CD38-daratumumab complexes from myeloma cells to monocytes and granulocytes by a process called trogocytosis [136]. In accordance with all these data, results from our group demonstrate in a preclinical model of acquired resistance to daratumumab that resistant cells express lower levels of CD38 than sensitive cells, both at mRNA and protein level (unpublished data). On the contrary, the in vitro continuous exposure of myeloma cells to effective concentrations of isatuximab does not result in a decrease of surface CD38 [116]. Moreover, preliminary data suggest that CD38 expression on primary MM cells is preserved during treatment with MOR202 [120]. It has also been described that both daratumumab and isatuximab induce internalization of CD38 [116,137], although a direct correlation between this event and the resistance to these mAbs has not been clearly established.

The overexpression of complement inhibitory proteins (CIPs) CD55 and CD59 at the time of disease progression, as compared to the levels before or during daratumumab treatment, suggests that there might be other potential mechanisms of acquired resistance to daratumumab [126]. However, in the same study authors observed that cell surface expression of those CIPs on MM cells prior to treatment is not associated with clinical response, suggesting that CIPs may not be useful as predictive biomarkers of response [126].

Finally, modifications in the expression of the adhesion molecule CD56 (NCAM1) have also been reported after treatment with daratumumab. In this sense, a flow cytometry analysis of 41 patients treated with daratumumab-based therapy revealed that CD56 expression was downregulated after treatment in non-responders versus responders [128]. In relation to this, Krejcik et al. observed that after daratumumab treatment, the downregulation of CD38 by trogocytosis was associated with reduced surface levels of some other membrane proteins, including CD56 [136]. Nevertheless, the downregulation of CD56 in this study was observed in both responders and non-responders, suggesting that more studies are needed to clarify if CD56 has a role in resistance to daratumumab [136].

### 4.3. Overcoming Drug Resistance

Since infra-expression of the target seems to be involved in resistance to mAbs, increasing its expression could resensitize, at least in part, myeloma cells to the drug. In this sense, augmentation of CD38 expression in myeloma cells induced by agents such as all-*trans* retinoic acid (ATRA), or panobinostat, improved the efficacy of daratumumab both in vitro and in vivo [125,138]. Moreover, ATRA also reduces the expression of the CIPs CD55 and CD59, contributing to improving daratumumab-mediated CDC [125]. In addition, considering the importance of immune-mediated cytotoxicity developed by mAbs, the combination of these mAbs with immune-activating drugs (e.g., IMiDs, CELMoDs, etc.) would potentiate and prolong their effects.

## 5. Conclusions

The development of resistance to therapeutic agents and the subsequent relapse is a general problem faced by most myeloma patients. Despite newer and more efficient therapies being tested in MM, preclinical studies show that myeloma cells manage their way to develop multiple and therapy-specific mechanisms of resistance; however, from the understanding of those mechanisms, therapeutic strategies have also been identified to overcome some of those resistance mechanisms. A lot of work is still to be done to confirm which of the identified mechanisms of resistance are the most prevalent on refractory myeloma patients and whether resensitization to a specific therapy is feasible in the clinical setting. Finally, there is also a great need for the identification of reliable and sensitive biomarkers of resistance, and ideally, predictive ones, so that in line with precision therapy for MM, more efficacious therapeutic strategies would be offered to patients trying to avoid the emergence of resistance.

## Figures and Tables

**Figure 1 cells-08-01432-f001:**
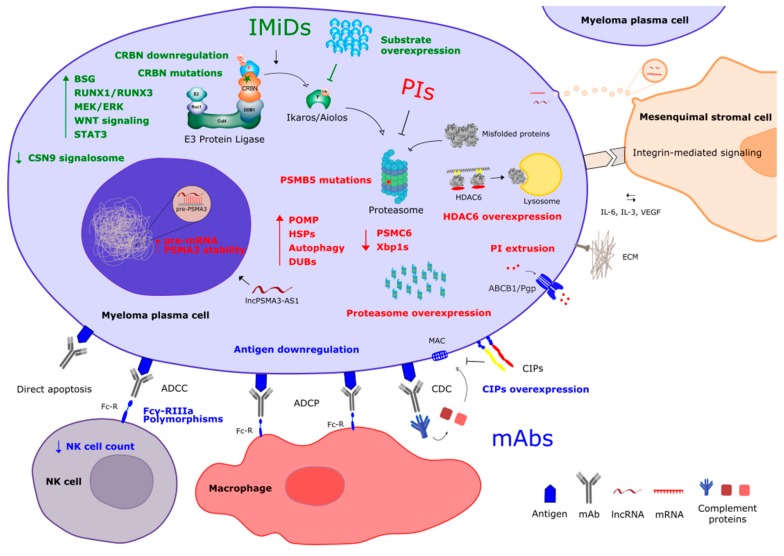
Schematic representation of main resistance mechanisms described to the present backbones in multiple myeloma (MM) treatment: proteasome inhibitors (PIs), immunomodulatory agents (IMiDs), and the more recently incorporated monoclonal antibodies (mAbs). Mechanisms of resistance to PIs contain lettering in red, to IMiDs in green, and to mAbs in blue.

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
