# Peer review of "Biological Background of Resistance to Current Standards of Care in Multiple Myeloma"

_cells, 2019, doi:10.3390/cells8111432_

Round 1

Reviewer 1 Report

This is a very well written review of the current knowledge on mechanisms of action and resistance mechanisms in MM. Some language proofs should be applied, though. 

I have only minor comments and suggestions:

1) a short summarizing paragraph on both clonal evolution and spatial heterogeneity as mechanisms of resistance would further strengthen the manuscript.

2) Figure 1 is very good, but would benefit from professional graphic design

3) the role of antioxidative capacity of MM cells in IMiD resistance, descripbed by Sebastian et al. could be added

4) line 88, what does "induces a fit" mean ?

5) line 152, "Relative to the role..." should be rephrased for better readability 

Reviewer 2 Report

Mogollon et al. described comprehensively the biological basis and mechanisms of drug resistance in multiple myeloma.  They focused on novel agents, proteasome inhibitors, IMIDs and mAbs.  So the title should be changed by specifying this.  

Comments

The review is well organized, and complete. the references are comprehensive and selected. The complexity of the biological mechanisms could be simplified in the parts containing more molecular aspects (given the audience would be also clinicians).

Page 5, mechanism of action of IMIDs. The MOA of IMIDS are discussed extensively, anyway I suggest to add also recent findings on the kinase CK1alpha, which is a target of Lenalidomide, since there are relevant papers on that describing a role in IFN pathway, signaling and autophagy in MM.
